# A dynamical anthrax toxin nanopore biosensor for high-fidelity single-peptide classification

Jennifer M. Colby[1][¤], Bryan A. Krantz[2]*

**1** Molecular Toxicology Graduate Program, University of California, Berkeley, California, United States of America, **2** Department of Microbial Pathogenesis, School of Dentistry, University of Maryland, Baltimore, Baltimore, United States of America

¤ Current address: Premier Biotech Labs, Minneapolis, Minnesota, United States of America
* bkrantz@umaryland.edu

## Abstract

Nanopore sensing holds the potential to revolutionize proteomics, yet current methods often rely on ensemble aggregation, where thousands of events must be statistically aggregated or averaged to identify a protein or peptide signature. While effective for pure samples, this aggregation strategy fails in complex, heterogeneous mixtures where the identity of individual molecules must be determined in real-time. Here, we demonstrate high-fidelity classification of peptides from single, individual translocation events, eliminating the need for ensemble averaging. This sensitivity is achieved using the anthrax toxin protective antigen (PA) nanopore. Unlike static pores used more generally, the PA pore's dynamic active-site clamps generate information-rich, multi-state signals. These clamps also enable the utility of high-affinity capture, permitting analysis at low nanomolar concentrations. We developed a machine learning framework that makes inferences on these dynamical multi-state signals and achieves ~91% accuracy on single events. This work establishes a framework for true single-molecule proteomics, enabling the resolution of complex mixtures that bulk aggregation methods cannot decipher.

## Author summary

Most nanopore sensors function like static molecular calipers, identifying molecules based solely on their size as they pass through a fixed channel. However, this approach often struggles to distinguish chemically similar molecules without averaging thousands of signals together. In this study, we introduce a fundamentally different approach by repurposing the anthrax toxin nanopore protein as a 'dynamic' biosensor. Unlike static pores, this biological machine possesses active moving parts that grab and interrogate molecules as they pass through. We show that these dynamic interactions generate complex, unique signal patterns—essentially a kinetic fingerprint—for every single molecule. Using a

**Data availability statement:** All experimental electrophysiological records and related source code are publicly available. The datasets used in this manuscript have been deposited in the Zenodo repository under the DOI: 10.5281/zenodo.16983789. The source code is maintained on a GitHub repository (https://github.com/bakrantz/Pept-Class).

**Funding:** This work was supported by the National Institutes of Health, Institute for Allergy and Infectious Disease (NIAID) funding (R01 AI077703 B.A.K.) The funders had no role in study design, data collection and analysis, decision to publish, or preparation of the manuscript.

specialized machine learning framework, we successfully decoded these patterns to identify specific peptides with over 90% accuracy from just a single event, rather than an average. We could even distinguish peptides that were identical in mass but differed only in their atomic arrangement. This work demonstrates that by using a sensor that moves and adapts, we can achieve high-precision, single-molecule identification. This capability is a critical step toward future diagnostic tools capable of analyzing complex biological mixtures, such as blood, where traditional averaging methods fail to resolve individual components.

## Introduction

The rapid and accurate detection and characterization of biomolecules, particularly peptides and proteins, represent a critical and often unmet challenge across diagnostics, drug discovery, and fundamental biological research. Peptide biomarkers, indicative of various disease states including heart disease, infectious diseases, and cancer [1–3], offer immense potential for timely and precise diagnosis. However, analyzing complex peptide mixtures *in situ*, often at low concentrations, demands analytical technologies with unparalleled sensitivity and specificity—capabilities that remain challenging for many conventional methods. Single-molecule nanopore biosensing provides a transformative approach [4], enabling the detection and characterization of individual molecules as they traverse a nanometer-scale pore by measuring picoamp-scale modulations in ionic current. Beyond simple presence/absence detection, nanopore technology holds the promise to revolutionize biopolymer analysis, including the ambitious goal of direct, high-throughput peptide and protein sequencing, a major frontier where robust, widely applicable methods are still lacking [5–7].

Nanopore biosensors consist of a membrane-embedded pore separating two electrolyte-filled compartments. Under an applied driving force (either a voltage or proton gradient), biomolecules are directed through the pore, generating unique current signatures dependent on their size, shape, and chemical properties. Biological protein nanopores, such as those formed by transmembrane proteins inserted into lipid bilayers, offer exquisite control over pore geometry and molecular interactions [8–14]. Single-channel recordings capture the dynamic, stochastic interactions of individual molecules with the nanopore in high-resolution ionic current traces. Extracting the maximum analytical information from these complex, high-dimensional translocation event streams is the central bottleneck preventing the full realization of nanopore technology's potential for analyzing complex biological samples.

To date, significant progress has been made in 'nanopore peptide profiling,' where proteins are digested and the resulting peptide flux is analyzed [15]. However, these approaches predominantly rely on ensemble fingerprinting: they aggregate statistics from thousands of translocation events to identify the parent protein. While powerful for verifying the identity of pure samples, this aggregation-based strategy faces a fundamental bottleneck when applied to realistic biological fluids or complex mixtures, where simultaneous signals from diverse analytes cannot be easily deconvolved. To

realize the full potential of nanopore sequencing and diagnostics, the field must move from ensemble averages to high-fidelity single-event, single-peptide classification.

Achieving this resolution requires more than a static measurement of excluded volume. Most nanopore platforms utilize structurally static pores, where the signal is limited to a simple blockade depth and dwell time. In contrast, we exploit the dynamical nature of the anthrax toxin protective antigen (PA) translocase [16]. By utilizing a pore that actively interrogates the substrate through distinct kinetic clamp site gating states, we generate a high-dimensional feature set for every single molecule. This allows us to move beyond the 'fingerprinting' of bulk samples and achieve >90% classification accuracy on individual translocation events, identifying subtle chemical differences—and even backbone stereoisomers—that static, aggregation-based methods miss.

Analyzing the massive, complex, and often noisy datasets generated by multi-state nanopore systems requires advanced computational approaches. Traditional manual or simple threshold-based analysis methods are fundamentally inadequate for extracting the full information content from nuanced multi-state kinetics or dissecting complex mixtures of analytes. Machine learning (ML) and deep learning (DL) offer powerful, modular, and adaptable tools uniquely suited to identify subtle patterns in translocation state sequences and correlate them with computed biophysical features [17–21]. While ML/DL has been applied to nanopore data, its application to analyzing real-world, multi-state peptide translocation data, either for precise peptide classification in mixed samples or for determination of sequence information, remains largely an underdeveloped area.

Here, we leverage the anthrax toxin PA nanopore to demonstrate high-fidelity peptide classification from individual translocation events. We develop a robust ML framework to decode the pore's complex, dynamical signals and prove its exquisite sensitivity by classifying a diverse set of guest-host peptides, including those differing only by backbone stereochemistry. While our previous recent work established the theoretical feasibility of this approach using 3-state translocation simulations [22], simulations cannot replicate the complex stochastic noise, 4-state gating, and lipid-protein interactions of a biological membrane. This work represents a real-world experimental realization while establishing a new class of dynamical nanopores for biosensing and providing a computationally efficient strategy for unlocking their information-rich data.

## Results

### Diverse guest-host peptide translocation events via PA nanopores

Previous investigations extensively characterized the broad ensemble properties and single-channel dynamics of the guest-host peptide series translocating through the anthrax toxin protective antigen (PA) nanopore [23,24]. Building upon this foundation, we sought to explore whether the intrinsic dynamic/kinetic properties of these peptides, as observed during single-channel translocation events, could serve as information-rich signatures for classification. Specifically, we aimed to assess if the PA nanopore, when coupled with powerful computational ML/DL methods, could reliably distinguish peptides based on subtle sequence differences that are challenging to discern by conventional analysis. This study thus assesses the PA nanopore's potential as a sophisticated peptide biosensor.

Relative to other protein nanopores commonly utilized in biosensing and nucleic acid sequencing, the PA nanopore (Fig 1A) possesses a significantly longer lumen. This architecture, however, features numerous active site clamps (e.g., α-clamp, ϕ-clamp, and charge clamp) and loop regions (e.g., 397-loop), specifically evolved to processively translocate large (~100 kDa) proteins, and, at nanomolar concentrations, shorter peptides. The 10-residue guest-host peptide series, with a general sequence of KKKKKXXSXX, was initially designed to systematically probe differences in binding, translocation, and dynamics based on the guest residue (X) (Fig 1B). Our guest-host panel included peptides with standard natural L-stereochemistry for guest residues Ala, Leu, Phe, Thr, Trp, and Tyr. To investigate the sensitivity of the nanopore to stereochemical variations, a seventh peptide, called guest-host TrpDL, was included; it shared the same amino acid sequence as guest-host Trp but featured an alternating pattern of D- and L-stereoisomers along its backbone.

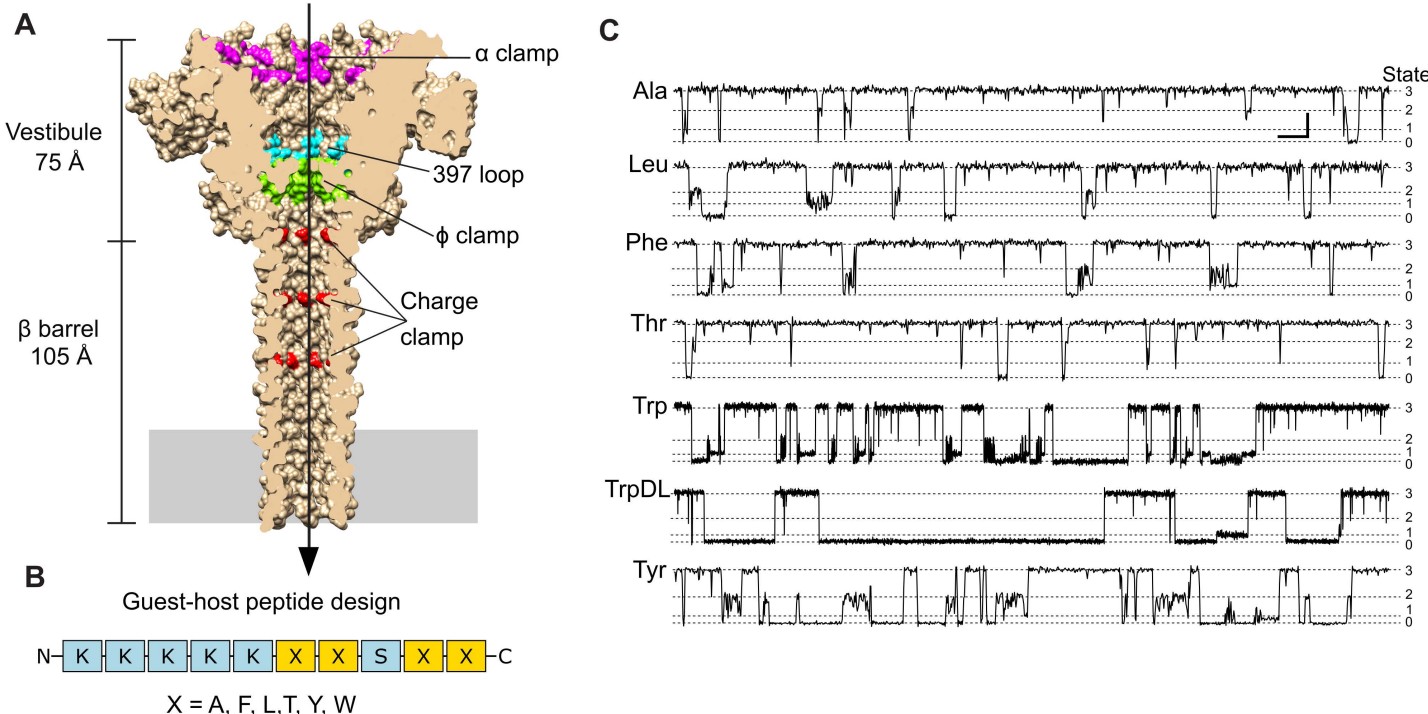

**Fig 1. PA nanopore peptide biosensor. (A)** Sagittal section of the PA nanopore (PDB: 3J9C) [8] rendered in Chimera [42] as a molecular surface. Peptide clamps and loop active sites are indicated: α clamp (magenta), 397-loop (cyan), φ clamp (green), and charge clamp (red). Narrowest point, at the φ clamp, has a luminal diameter of 6 Å. Direction of translocation from *cis* to *trans* is indicated by an arrow. Membrane bilayer position is indicated with a solid gray rectangle. **(B)** Guest-host peptide design schematic for the 10-residue guest-host peptide with general sequence, KKKKKXXSXX, with guest residue (X). **(C)** Representative current versus time records of guest-host peptide translocations carried out at 70 mV (*cis* positive) in symmetric succinate buffer, 100 mM KCl, pH 5.6. To the left are guest-host peptide names. In general, this nanopore-peptide system populates multiple discrete conductance state intermediates (indicated by dashed lines), which are enumerated by state on the far right: fully blocked (state 0), partially blocked intermediates (state 1 and state 2), and fully open (state 3). Scalebar at upper right for guest-host Ala, Leu, Phe, Thr, and Tyr peptides represents 2 pA by 100 ms. For guest-host Trp and TrpDL peptides, the scalebar represents 2 pA by 500 ms due to their characteristically longer events.

The N-terminal five-lysine (K5) leader sequence was selected for two critical biophysical reasons. First, the PA nanopore is cation-selective [25]; the highly positively charged K5 tail ensures rapid electrophoretic capture and entry into the pore lumen under the applied positive membrane potential. Second, this leader sequence serves as a kinetic brake, preventing the short peptide from translocating too rapidly for resolution, effectively mimicking the charged N-terminal leader sequences required for the translocation of native enzymatic factors of the toxin [26].

For this assessment of the PA nanopore as a biosensor platform, we focused on collecting extensive single-channel translocation event streams via planar lipid bilayer electrophysiology. The data acquisition and processing workflow is shown in S1A Fig. All analyses presented herein utilized data acquired under a consistent 70 mV driving force (*cis* positive). This potential strongly favors full translocation events for these peptides [23,24] and longer proteins [27,28], which is critical for consistent feature extraction as it ensures the peptide fully interacts with the entire nanopore rather than more superficially engaging the entrance, as might occur much more often at lower potentials. Furthermore, the signal-to-noise ratio is inherently higher at larger potentials, providing additional support for this selection. A comprehensive breakdown of the total recording times per peptide in our complete dataset is provided in S1 Table.

Across samples of translocation event streams for the seven guest-host peptide classes, four discrete conductance states are consistently observed, which are enumerated as states 0–3 (Fig 1C). These translocation events can be

reliably observed because the wild-type PA nanopore in the absence of peptide at pH 5.6 is relatively stable with only periodic short spikes attributed to ϕ clamp 'wetting' and 'dewetting' (S2 Fig) [29]. The observed translocation event dynamics, as characterized by current blockade patterns and durations, cover a broad range. Qualitatively, events for guest-host Trp and guest-host TrpDL exhibit noticeably longer durations compared to the other five peptides. In contrast, peptides with smaller guest residues, such as guest-host Ala and guest-host Thr, display rapid dynamics that are often difficult to distinguish reliably by visual inspection alone. For peptides like guest-host Leu and guest-host Phe, which are similar on hydrophobicity scales, subtle visual differences in their translocation dynamics exist but are similarly challenging to resolve. The aromatic guest residues Phe and Tyr present event lengths and flickering dynamics that are visually analogous and difficult to discriminate. Therefore, these qualitative observations, particularly the subtle or complex nature of visual distinctions between peptide classes, directly suggest that advanced ML/DL methods can be effectively exploited to robustly classify these peptides, even from individual single-channel translocation events.

## Discriminatory potential of engineered features is event length dependent

Achieving robust peptide classification from individual translocation events necessitated a comprehensive and generalized feature engineering approach. To ensure high-fidelity state assignments critical for feature extraction, raw current records were state-labeled using the 'Single-Channel Search' routine in CLAMPFIT, providing expert-validated assignments. For our specific four-state event streams [24], a consistent enumeration scheme was used: state 0 for fully blocked, states 1 and 2 for intermediate blockades (closest to state 0 and state 3, respectively), and state 3 for the fully open state (Fig 1C). From these labeled records, both conductance state sequences and scaled current sequences were segmented for each translocation event, from which a rich feature set (including scalar, vector, and matrix features) was calculated (see Materials and Methods) (S1B Fig and S2 Table). While the segmentation process offers filtering and baseline correction capabilities, these were not employed for the current datasets. However, we critically explored the impact of minimum event duration as a preprocessing filter. Physically, extremely short events largely represent superficial vestibule collisions culminating in retro-translocation back to the *cis* compartment, or transient 'wetting/dewetting' transitions of the hydrophobic ϕ clamp (S2 Fig) [29]. These non-productive events lack the full sequence of kinetic transitions required for identification. In contrast, longer events capture the dynamic interaction between the peptide and the clamp sites, effectively providing 'multiple reads' of the analyte as it negotiates the energy landscape. While a more relaxed cutoff (e.g., 12.5 ms) still yielded robust performance, we selected the 20 ms threshold to maximize signal fidelity, ensuring the classifier relies on high-quality, information-rich translocation signatures rather than noisy, incomplete sampling. The main practical downside to using a minimum event duration filter is higher filtering values for this parameter could remove large numbers of translocation events from the dataset (S3 Table).

To qualitatively and quantitatively assess the impact of different minimum event duration thresholds on the discriminative power of the extracted feature sets, Uniform Manifold Approximation and Projection (UMAP) dimensionality reduction was employed to generate 2D cluster representations. For events filtered at a minimum duration of 5 ms, UMAP analysis (Fig 2A) resulted in poor clustering performance. Visually, peptide classes remained largely intermixed, with few well-structured, distinct clusters and numerous 'stray' points dispersed throughout the 2D projection. In contrast, increasing the minimum event-length filter to 20 ms visibly improved clustering in the UMAP analysis (Fig 2B). While some intermixing persisted and a smaller fraction of stray points remained, distinct clusters became apparent for several peptides, notably Phe, Thr, Trp, and Tyr. Interestingly, despite its strong classification performance in subsequent ML/DL models (as shown later), the events for guest-host TrpDL in the 20 ms UMAP embedding presented as several smaller, somewhat dispersed clusters, suggesting inherent sub-populations or more complex relationships not fully captured by the 2D projection. To quantify these visual observations, clustering metrics Adjusted Rand Index (ARI) and Normalized Mutual Information (NMI) were computed by applying K-Means clustering (K = 7, for seven peptide classes) to the UMAP embeddings and comparing the resulting clusters to the known peptide labels. For data filtered at 5 ms, the ARI was 0.0356 and

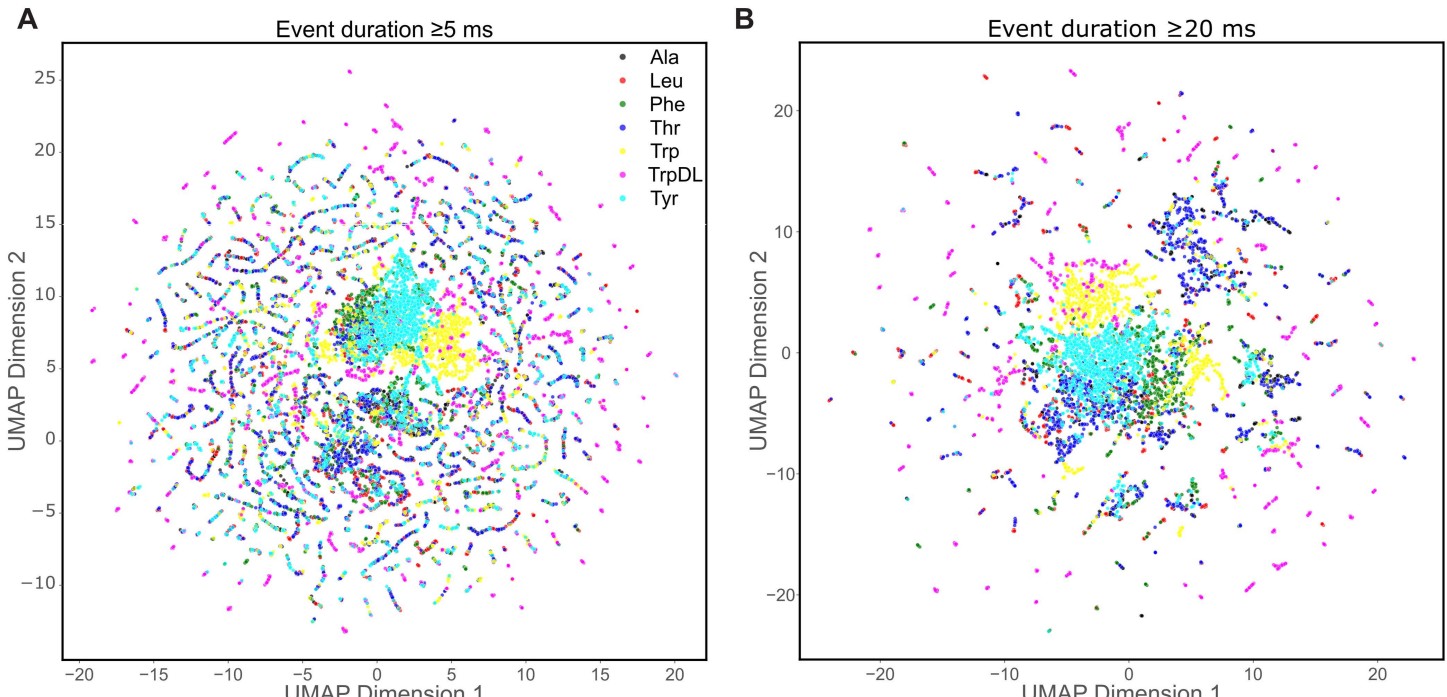

**Fig 2. Discriminative potential of extracted features at different minimum event durations.** UMAP clustering analysis of event-level features. Data points represent individual translocation events, colored by their corresponding guest-host peptide identity: Ala (black), Leu (red), Phe (green), Thr (blue), Trp (yellow), TrpDL (magenta), and Tyr (cyan). **(A)** UMAP embedding of events filtered at a minimum duration of 5 ms (n_neighbors = 15, min_dist = 0.5). Peptide classes are more intermixed with less clear separation, indicating more limited discriminative power of features for very short events (ARI: 0.0356; NMI: 0.0402). **(B)** UMAP embedding of events filtered at a minimum duration of 20 ms (n_neighbors = 15, min_dist = 0.5). In contrast, events filtered at 20 ms show visibly improved clustering for several guest-host peptides (e.g., Phe, Thr, Trp, Tyr), albeit TrpDL is more spread out and peripheral. While some scattered points remain, suggesting inherent event variability or limitations of the 2D projection, the overall separation is markedly enhanced (ARI: 0.0887; NMI: 0.1282).

NMI was 0.0402. However, for data filtered at 20 ms, the ARI increased to 0.0887 and the NMI to 0.1282. The substantially higher ARI and NMI values for the 20 ms filtered data strongly indicate a greater alignment between UMAP-derived clusters and true peptide identities at longer event durations. These results collectively demonstrate that the engineered feature sets gain significant discriminative power by excluding very short events, albeit with the inherent trade-off of reducing the total number of analyzed events (S3 Table). It should also be noted that we systematically filtered events <20 ms not to reduce dataset complexity, but to satisfy the Nyquist-Shannon sampling requirements of the φ-clamp dynamics. Events shorter than the characteristic relaxation time of the clamp do not contain sufficient kinetic information for classification and may represent simpler vestibule collisions rather than full dynamic translocation.

## Performance of supervised DL classification models

The core design approach for DL-based peptide classification from individual translocation events involved a branched, dual-input neural network architecture (Figs 3A and S1C) [22]. In this configuration, either the conductance state sequences (S) or the raw current sequences (C) from translocation events served as input to a multi-layered CNN or TCN branch. Simultaneously, the corresponding event-level features (F), extracted during preprocessing, were fed into a separate fully connected (Dense) network. The outputs of these two parallel branches were then concatenated before leading to a final classification layer. Based on this design pattern, three distinct configurations were assessed: TCN-Dense

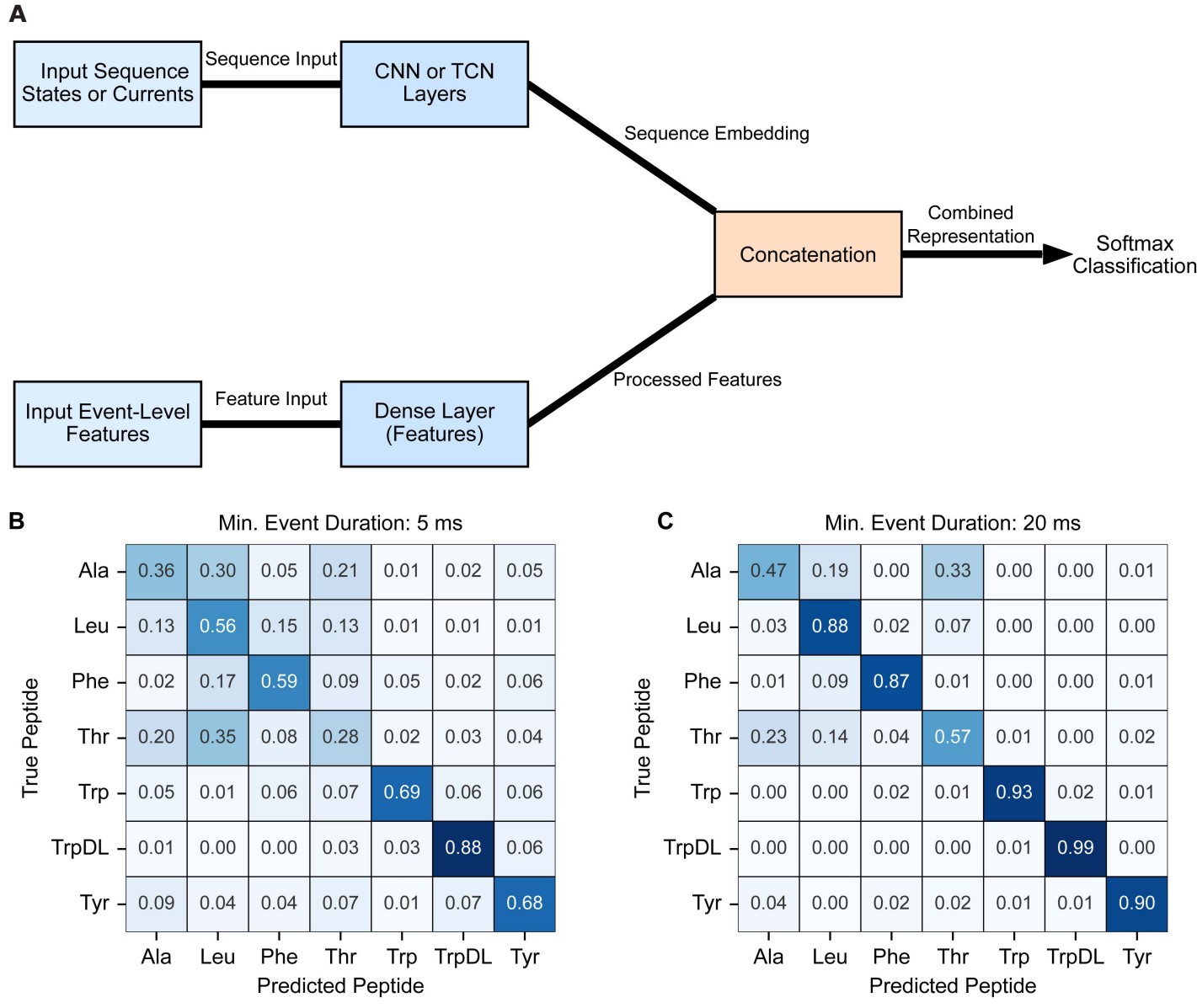

**Fig 3. DL-based classification of guest-host peptide translocation events. (A)** General dual-input neural network architecture as a block diagram illustrating the common design pattern for the DL classifiers. Sequence data (either conductance states (S) or raw current (C)) are processed by a multi-layered CNN or TCN branch. Concurrently, event-level features (F) are processed by a fully connected (Dense) branch. The outputs of these two branches are concatenated and fed into a final Dense layer for classification. This architecture was the basis for the TCN-Dense (S+F), CNN-Dense (S+F), and CNN-Dense (C+F) models. **(B)** Normalized confusion matrix displaying the per-class prediction accuracy of the CNN-Dense (C+F) model on the test set for a minimum event duration of 5 ms. Rows represent true peptide labels, and columns represent predicted labels. Values indicate the proportion of events from a given true class that were predicted as each class. **(C)** Normalized confusion matrix for CNN-Dense (C+F) at 20 ms minimum event duration, which is plotted as in panel B. Note the improved diagonal elements (correct predictions) and reduced off-diagonal elements (misclassifications) compared to the shorter minimum event duration predictions in panel B. These matrices are plotted with absolute event counts in S3 Fig.

(S+F), CNN-Dense (S+F), and CNN-Dense (C+F). Given that the discriminative power of the engineered features was shown to be dependent on the minimum event duration parameter during preprocessing (Fig 2), these models were subjected to an initial, single-replicate performance scan across a range of minimum event duration values (5, 7.5, 10,

12.5, 15, and 20 ms) (S4 Table). This preliminary scan revealed a consistent trend across all DL models: increasing the minimum event duration to exclude shorter, less information-rich events significantly improved classification performance. Finalized, comprehensive performance metrics (mean ± standard deviation, N = 5 independent training runs) were compiled for these models at the two extreme minimum event duration cutoffs (5 and 20 ms) (Table 1).

The TCN-Dense (S + F) model, a capable architecture previously developed for event-level classification of simulated peptides [22], served as a baseline in our evaluation. This model, trained on translocation event state sequences and corresponding event-level features, showed a modest average accuracy (0.6287 (±0.0157) at 20 ms cutoff) and generally followed the trend of improved predictions with increasing event length (Table 1). The CNN-Dense (S + F) model, likewise trained on state sequences and event-level features, surprisingly achieved slightly better performance (e.g., 0.7155 (±0.0095) accuracy at 20 ms cutoff, N = 5), while being less computationally demanding (Table 1). Critically, the CNN-Dense (C + F) model, which was trained directly on the scaled current sequences and event-level features, emerged as the most robust DL model. Its superior performance (e.g., 0.7857 (±0.0116) accuracy at 20 ms cutoff, N = 5) strongly suggests that the raw current sequences of translocation events possess more discriminative information than the more abstract conductance state sequences (Table 1).

Detailed confusion matrices for the CNN-Dense (C + F) model at both 5 ms and 20 ms minimum event durations (Fig 3B and 3C) further elucidate class-by-class prediction quality. These reveal that the aromatic guest-host peptides (Phe, Trp, TrpDL, and Tyr) were consistently predicted with high fidelity, even considering their nuanced chemical differences. Notably, the models demonstrated the nanopore's ability to discriminate between guest-host Trp and guest-host TrpDL, which differ only in backbone stereochemistry. This finding strongly suggests that peptide backbone conformational dynamics, previously reported as key biophysical properties detectable by the PA nanopore [24,30], are indeed unique features leveraged by the classifier. Conversely, consistent misclassification was observed between guest-host Ala and guest-host Thr. Their rapid translocation events exhibited similar dynamics and consequently occupied overlapping feature space, reflecting the subtle chemical differences between these two residues. For such cases, the analysis consistently indicated that increasing the minimum event duration parameter, thereby focusing the classification models on longer, more information-rich events, represents the best strategy for improving classification accuracy for these challenging cases.

**Table 1. Model performance metrics at two minimum event duration extremes[1].**

| Model[2] | Metric[3] | Minimum event duration | |
|---|---|---|---|
| | | 5 ms | 20 ms |
| XGBoost (F) | Accuracy | 0.5419 (±0.0066) | 0.9112 (±0.0069) |
| XGBoost (S + F) | | 0.5334 (±0.005) | 0.9047 (±0.0064) |
| CNN-Dense (C + F) | | 0.5511 (±0.0141) | 0.7857 (±0.0116) |
| CNN-Dense (S + F) | | 0.4201 (±0.0069) | 0.7155 (±0.0095) |
| TCN-Dense (S + F) | | 0.3821 (±0.0144) | 0.6287 (±0.0157) |
| XGBoost (F) | F1-score | 0.5371 (±0.0056) | 0.9109 (±0.0068) |
| XGBoost (S + F) | | 0.5255 (±0.0069) | 0.9044 (±0.0063) |
| CNN-Dense (C + F) | | 0.5517 (±0.0146) | 0.7845 (±0.0108) |
| CNN-Dense (S + F) | | 0.4265 (±0.0071) | 0.7112 (±0.0071) |
| TCN-Dense (S + F) | | 0.3948 (±0.0157) | 0.6296 (±0.0166) |

[1]Performance based on test set evaluation metrics of different ML/DL classification models at different minimum event durations. Values are means and std. dev. (N = 5).

[2]Models are named as defined in the text.

[3]Metric names are abbreviated and refer to overall accuracy and macro-averaged F1-score.

## Performance of ML and ML/DL-hybrid classifiers

Beyond DL approaches, we investigated the performance of tree-based machine learning classifiers. Specifically, XGBoost, a highly efficient ensemble learning method leveraging gradient-boosted decision trees [31], was employed using only the pre-extracted event-level feature set—referred to as XGBoost (F). Like the DL models, a preliminary single-replicate scan across the same range of minimum event durations (5–20 ms) revealed a consistent trend: classification metrics significantly improved as shorter, lower-information events were progressively removed from the training data (S4 Table). At a minimum event duration of 20 ms, the XGBoost (F) model achieved exceptionally high performance, with an average overall accuracy of 0.9112 (±0.0069) (N = 5) (Table 1). A detailed normalized confusion matrix for this configuration (Fig 4) illustrates the robust per-class prediction quality. Consistent with observations from other models, guest-host Ala and guest-host Thr remained the weakest performing classes, exhibiting some low-level confusion (Fig 4). However, their F1-scores were still notably high, exceeding 0.8. A significant practical advantage of XGBoost over the DL-based approaches was its substantial computational efficiency during both model training and inference.

We also explored a hybrid ML/DL model architecture to assess if sequence-derived embeddings could enhance feature-based classification by XGBoost [31]. This hybrid approach XGBoost (S + F) utilized a CNN to generate embeddings from the state sequences, which were then merged with the event-level feature set, and this combined vector was subsequently used to train an XGBoost classifier. While this hybrid model also demonstrated increasing performance with longer minimum event durations, it did not outperform the XGBoost (F) model acting solely on the event-level feature set (Table 1). This indicates that, for our nanopore/peptide system, the addition of CNN-derived embeddings from state sequences did not provide significant additional discriminative information when combined with the already comprehensive engineered event-level features. In conclusion, for our nanopore and guest-host peptide system, the combination of

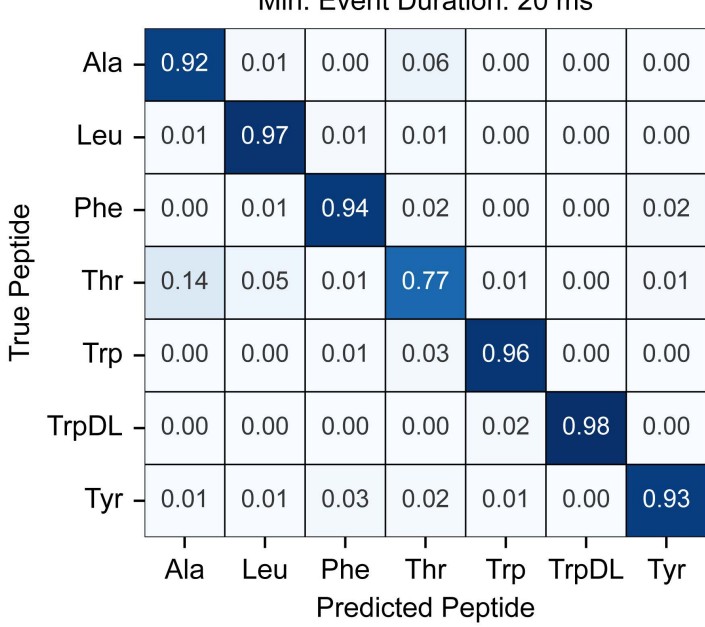

**Fig 4. High-performance tree-based classification of translocation events.** Normalized confusion matrix for XGBoost classification using the event-level feature set. This matrix represents the best-performing XGBoost (F) model at a minimum event duration of 20 ms. Rows represent true peptide labels, and columns represent predicted labels. Values indicate the proportion of events from a given true class that were predicted as each class. Despite the model's overall high performance, a very moderate level of misclassification is observed between the chemically similar guest-host Ala and guest-host Thr classes. This matrix is plotted with absolute event counts in S4 Fig.

carefully engineered features (particularly those derived from longer events) with a robust tree-based model like XGBoost (F) achieved superior or highly comparable classification performance with significantly greater computational efficiency than the tested deep learning sequence models.

## Discussion

### Dynamical clamp sites enhance PA's performance as a nanopore biosensor

Currently, most nanopore platforms have relied on structurally static pores—both protein-based and solid-state nanopore systems. In this limit, the pore acts as a passive tunnel, and sensing relies on measuring the analyte's excluded volume, analogous to using a fixed size molecular caliper. In contrast, dynamical systems, such as dedicated protein translocases, have not been exploited for nanopore biosensing. These proteins possess active-site clamps and loops, like the φ clamp [14], that undergo conformational changes, offering a new mechanism for sensing based on using dynamic active-site interactions as well as analyte size [32]. The PA nanopore highlighted here is the first-in-class example of such a dynamical biosensor. Dilated and constricted clamp states with distinct peptide interaction thermodynamics and kinetics are apparent in single-channel electrophysiology [24,30,32,33]. The population of these discrete conformational states during peptide translocation enables multiple measures of peptide identity to be made during a single translocation event. ML/DL models can interpret these dynamical, albeit complex, signals faithfully, as we have shown here. Finally, dedicated protein translocases, which possess multiple high-affinity clamps (like the φ clamp), allow biosensing to be performed at low nanomolar concentrations, a significant sensitivity advantage over other static nanopores.

### Exquisite sensitivity and broad discriminatory power of PA nanopore

The models consistently demonstrated the ability to discriminate between peptides differing only in backbone stereochemistry (e.g., guest-host Trp and TrpDL), highlighting the remarkable intrinsic sensitivity of the PA nanopore to subtle conformational and interaction dynamics (Figs 3C and 4). Previous biophysical characterizations of the PA nanopore revealed altered peptide and clamp dynamics [24] and helix-coil transition dynamics [30], when only stereochemistry was manipulated. This conformational biosensing capability is likely somewhat unique to this nanopore system. Furthermore, the discriminatory power extended to other chemically subtle differences between peptides. Guest-host Phe and guest-host Tyr were well classified despite differing only by a hydroxyl group. Static nanopore architectures, which are configured as fixed molecular calipers, may not be as versed in sensing small changes in peptide size. Guest-host Phe and guest-host Leu, despite having similar hydrophobicity, were distinguishable by this pore and computational approach. Early work on model compound binding to φ-clamp site (F427) showed enhanced binding of aromatic compounds, suggesting ring-ring interactions are made to the phenyl groups [14]. Overall, these observations underscore the nanopore's ability to discern fine chemical, structural, and conformational nuances beyond what might be expected.

Another distinct advantage of the PA nanopore is its high-affinity capture mechanism. While static nanopores typically require micromolar analyte concentrations to achieve practical event frequencies, the PA φ-clamp actively recruits peptides, enabling robust data acquisition at low nanomolar (5–20 nM) concentrations. Operating in this single-molecule regime is advantageous not only for sensitivity but also for data fidelity. While prior ensemble studies have noted concentration-dependent allosteric effects for specific substrates at high concentrations (e.g., guest-host Trp) [23], the low nanomolar concentrations employed here minimize concurrent occupancy and cooperative phenomena. Thus, within this operational window, the classification relies on the intrinsic kinetic fingerprint of individual translocation events, independent of bulk high concentration effects.

### Decoding the complex signals of a dynamical pore

A key breakthrough of this platform is the ability to classify peptides from individual translocation events, rather than relying on ensemble-averaged features. However, the same dynamical interactions that provide the pore's exquisite sensitivity

also generate complex, multi-state signals that can be noisy and difficult to interpret. We therefore systematically compared different computational strategies to find the most robust method for decoding these signals.

We found that while end-to-end deep learning models (e.g., CNN-Dense (C + F)) performed reasonably (achieving ~79% accuracy), a feature-based approach was demonstrably superior. Our engineered, event-level features—which distill the complex biophysical information from the multi-state current traces—proved highly discriminative. The XGBoost (F) model, relying only on these features, achieved the highest classification accuracy (~91%), outperforming all other architectures. This result establishes that, for this system, the engineered features capture the discriminative information more directly and robustly than the DL models could from the raw current sequences.

Another key insight was the critical role of event duration. We found that systematically filtering out short events (e.g., < 20 ms) dramatically improved classification performance across all models, an observation confirmed visually by UMAP. This indicates that longer translocation events carry more stable and discriminative information, likely because they represent more complete interactions with the pore's multiple clamp sites.

It is important to note the limitation of this feature-based approach: our feature set was optimized for four-state peptides. While this traditional ML strategy was superior for our well-characterized system, DL models will likely be essential for more complex, heterogeneous data—such as classifying peptides that exhibit a variable number of conductance states during translocation.

However, this strict requirement for distinct, four-state kinetic signatures serves a critical translational function. In complex clinical matrices (e.g., serum or lysates), non-specific background interactions often manifest as transient, featureless blockades that mimic biological noise. By optimizing for high-dimensional kinetic fingerprints, the algorithm effectively acts as a specificity filter, distinguishing true peptide analytes from these non-specific background events that lack the defined multi-state trajectory.

While Hidden Markov Models (HMM) are the standard for idealized, discrete-state ion channel data, they assume that the signal transitions between stable, memoryless states. However, the PA nanopore signal exhibits rapid, non-Markovian fluctuations and 'flickering' dynamics within the blocked states that contain high-frequency spectral information. Our DL (CNN/TCN) and ML (XGBoost) approaches were chosen specifically to capture these fine-grained 'kinetic fingerprints' directly from the current and states time series, extracting high-dimensional features that idealized HMM state-fitting may smooth over or discard.

## Challenges and future opportunities

Broadening this platform to the wider proteome will require navigating diverse physicochemical properties beyond the cationic peptides tested here. Previous studies have established that zwitterionic sequences can thread and translocate effectively under a proton gradient [34]. While dense stretches of anionic charge can slow translocation due to the pore's cation-selectivity, they do not strictly preclude it [13,34]; as pointed out in the earliest models [27], the acidic operating pH (5.6) facilitates the protonation of acidic side chains, neutralizing charge-charge repulsion and permitting transport. Highly hydrophobic peptides present a distinct challenge; beyond inherent solubility issues, they would likely stall within the hydrophobic φ clamp, resisting re-solvation and requiring higher driving forces or solubilizing leader sequences to traverse the pore.

The most challenging pair of peptides to classify here were guest-host Ala and guest-host Thr in all tested models. The current best strategy to reduce their misclassification and confusion is to increase the minimum event duration to filter out lower quality short events. Additionally, while we report binary classification accuracy here for rigorous benchmarking, the XGBoost algorithm is inherently probabilistic. In a future clinical deployment, the model's output probabilities (e.g., Softmax) would be utilized to flag and discard low-confidence events (e.g., probability < 0.8). This confidence gating would further enhance the effective specificity of the system beyond the raw accuracy reported here, ensuring that only high-certainty assignments are used for diagnosis.

The wild-type PA nanopore used here may not be perfectly suited to distinguish all chemistries. Therefore, a multiplexed approach can be advantageous, where a variant, orthogonal PA nanopore (e.g., a ϕ-clamp mutant, like F427A) is used [35]. Moreover, based on functional translocation studies this F427A variant would be considered a rational engineering of the pore's dynamical active site ϕ clamp [24,30,32,36]. This approach may offer better discrimination between these two challenging peptides. Having an alternate nanopore may offer orthogonal discriminatory information to augment inference.

While this study focuses on *single-pass* classification, the stochastic nature of single-molecule sensing offers a path to even higher accuracy via consensus sequencing. Similar to metrics used in commercial nanopore DNA sequencing, aggregating statistics from multiple independent events of the same analyte can exponentially increase classification confidence. Future implementations could also employ 'flossing' techniques—mechanically trapping and moving a single peptide back and forth through the pore—to generate multiple reads of a single molecule, allowing majority voting or Bayesian updates to push accuracy from ~90% to near certainty.

While this study focused on classification, insights gained here (feature engineering, event-length optimization, and analyte sensitivity) lay the groundwork for more ambitious goals like direct peptide sequencing. Neural networks are likely essential for this task. For example, a sequence-to-sequence (current-to-peptide sequence) architecture could be a powerful avenue to pursue. Beyond refining classification for known peptides, future work will also explore the generalizability of our approach to wider, chemically diverse peptide sets, complex mixtures, and even different nanopore architectures. Furthermore, the integration of more advanced feature extraction techniques, potentially leveraging deep learning for automated feature engineering or multi-modal data fusion, could further unlock unprecedented performance beyond our current methods. Finally, translating these advancements to real-world applications will necessitate addressing practical challenges inherent in complex biological matrices, such as sample preparation, signal-to-noise optimization, and high-throughput integration. Ultimately, by persistently refining both the nanopore biosensor and the computational intelligence that deciphers its signals, we move closer to a future where rapid, single-molecule proteomic analysis becomes a routine and transformative reality.

## Materials and methods

### Nanopore and peptides

Monomeric 83-kDa PA ($PA_{83}$) preprotein and the oligomerized heptameric form ($PA_7$) were prepared as described [37]. $PA_{83}$ monomer was overexpressed in *Escherichia coli* BL21(DE3), using a pET22b plasmid, which directs expression to the periplasm. Cell cultures were grown at 37 °C in a custom 5 L fermentor using ECPM1 growth media [38], which was supplemented with carbenicillin (50 mg/L). Once reaching an $OD_{600}$ of 3–10, the cultures were then induced with 1 mM isopropyl β-d-thiogalactopyranoside for ~3 h at 30 °C. $PA_{83}$ was released from the periplasm by resuspending pelleted cells on ice using a wire whisk with 1 L of hypertonic sucrose buffer (20% sucrose, 20 mM Tris-Cl, 0.5 mM EDTA, pH 8) followed by osmotic shock of centrifuged/pelleted cells using a wire whisk in 1 L of hypotonic solution (5 mM $MgCl_2$). Released $PA_{83}$, isolated after centrifugation to remove cellular debris, was purified on Q-Sepharose anion-exchange chromatography in 20 mM Tris-Cl, pH 8.0 by binding and then eluting with a linear salt gradient using 20 mM Tris-Cl, pH 8.0 with 1 M NaCl.

To make $PA_7$ prepore oligomers, purified $PA_{83}$ at a concentration of 1 mg/ml was treated with trypsin (1:1000 wt/wt trypsin:PA) for 30 min at room temperature to form nicked PA. Trypsin was subsequently inhibited with soybean trypsin inhibitor at 1:100 dilution (wt/wt soybean trypsin inhibitor:PA). Nicked PA was applied to Q-Sepharose to then isolate the $PA_7$; oligomer was bound to the column in 20 mM Tris-chloride, pH 8.0 and eluted by a linear salt gradient using 20 mM Tris-Cl, 1 M NaCl, pH 8.0. $PA_7$ was concentrated and frozen in small aliquots to maintain reproducible nanopore insertion activity in planar bilayer experiments.

Ten-residue guest-host peptides of the general sequence, KKKKKXXSXX, where X = A, L, F, T, W, and Y, were synthesized with standard L amino acids to >95% purity as indicated [23,24] (Elim Biopharmaceuticals). One stereochemical variant of X = W (abbreviated TrpDL) was produced, where instead of synthesizing the peptide with uniform L amino acids, an alternating pattern of D and L amino acids was used [24].

### Single-channel electrophysiology

Planar lipid bilayer currents were recorded using an Axopatch 200B amplifier interfaced by a Digidata 1440A acquisition system (Molecular Devices) [24,28,37]. Membranes were formed by painting across a 50-μm aperture of a 1-mL white Delrin cup with 3% (wt/vol) 1,2-diphytanoyl-*sn*-glycero-3-phosphocholine (Avanti Polar Lipids) in *n*-decane. The *cis* (side to which the PA$_7$ is added) and *trans* chambers were bathed in symmetric single-channel buffer (SCB: 100 mM KCl, 1 mM EDTA, 10 mM succinic acid, pH 5.60). Recordings were acquired at 400–600 Hz using PCLAMP10. The applied voltage is defined as $\Delta\psi = \psi_{cis} - \psi_{trans}$ (where $\psi_{trans}$ was set to 0 mV).

Single-channel recordings of the guest-host peptide translocations via the PA nanopore were carried out as described [24] with some slight differences. A single PA channel was inserted into a painted bilayer at a $\Delta\psi$ of 20–30 mV by adding ~2 pM of PA$_7$ (freshly diluted from a 2-μM stock) to the *cis* side of the membrane. The oligomer converts to the nanopore state by inserting into the membrane in an oriented manner. Once a single channel inserted, the *cis* chamber was perfused by fresh SCB to remove excess uninserted PA$_7$. Then the desired peptide analyte was added to the *cis* chamber at 5–20 nM. Translocation data were acquired by stepping the applied $\Delta\psi$ to a higher positive value and collecting recordings of the translocation event stream for up to several minutes.

Minor processing as well as conductance state labeling of the raw single-channel event stream recordings was subsequently performed. Rare transient out-of-range current spikes, insertion of second channels, and inactivated channels were removed by a 'force values' routine in CLAMPFIT. Some translocation recordings that were acquired at 500 and 600 Hz were downsampled to 400 Hz by decimation in Python using the scipy.signal library [39]. 400 Hz was chosen to maximize the data volume at a consistent time step for ML/DL-based peptide classification. Four discrete conductance states were then detected in these recordings using single-channel event detection in CLAMPFIT. We noted that during event detection the shortest translocations consisting of a single time step (2.5 ms at 400 Hz) were identified by CLAMP-FIT as being two time point long events (5 ms). By convention during event detection in CLAMPFIT, the fully blocked peptide-bound state was state 0, the intermediate closest to the fully blocked state was state 1, the intermediate closest to the open state was state 2, and the open state was state 3. Start and stop times of all detected events were used to label the state of each time point in the raw current recordings, producing a three-column CSV file of the stream with columns labeled as 'time', 'current', and 'state'. All labeled CSV stream files for the seven peptides were entered into a local annotated peptide database to aid in *in situ* loading/preprocessing for each tested ML/DL model (described below).

### Hardware and software used for ML/DL

Anaconda was used to create a Python 3.10.16 environment, where TensorFlow (2.16.2) [40], Keras TCN (3.5.6) [41], XGBoost (3.0.0) [31], and other standard modules were installed. The hardware used in training ML/DL peptide classification models was a 2025 MacBook Pro with M4 Apple Silicon and 24 GB of RAM. GPU cores were utilized during training by installing tensorflow-metal. All source code is available at GitHub (https://github.com/bakrantz/Pept-Class).

### Preprocessing, translocation event segmentation, and feature extraction

Raw translocation event streams were preprocessed prior to segmentation and feature extraction (S1B Fig) following a procedure developed on simulated translocation event streams [22]. While the segmentation core can apply low- and high-pass filtering and baseline correction, these filters/corrections were not applied to the experimental guest-host peptide translocation datasets presented in this study. The most critical preprocessing parameter was the minimum event

duration, which served as an effective filter for excluding very short-duration events. This parameter was systematically varied in the range of 5–20 ms to assess its empirical impact on downstream ML/DL classification performance.

Following the application of these processing parameters, the state-labeled raw event streams were segmented into individual translocation events [22]. Each event was defined as initiating when the current changed from the fully open state (state 3, corresponding to baseline current) to any peptide-bound state (state 0, 1, or 2) and terminating when the current returned to the open state. From these segmented events, both raw current sequences and corresponding state sequences were extracted.

A comprehensive set of event-level features was then computed from these sequences using a custom segmentation core [22]. This core maintains a generalizable framework to process peptide translocation events from systems exhibiting diverse mechanisms and an arbitrary number of states. These features were initially categorized into scalar, vector, and matrix data structures, with values in vector and matrix features being state or transition enumerated. Scalar features included: Shannon entropy of state sequence, event duration, number of transitions, time of the first transition, and total number of states visited during the event. Vector features included: observed conductance state Boolean, observed conductance levels, probability of residing in each state, and longest dwell time in each state. Matrix features included: average dwell time for specific state-to-state transitions, variance of dwell time for transitions, and ratio of probabilities between states. The descriptions and dimensions of this feature set are in S2 Table.

While the segmentation core also supports the extraction of global features that have yielded high-quality classification results in simulated datasets (e.g., > 0.99 accuracy) [22], these were not utilized in the classifications of the experimental guest-host datasets presented here, as our focus was solely on the more challenging and practical task of individual event-level classification.

For downstream classification, all matrix features were flattened into one-dimensional arrays and appended with the vector and scalar features to form a single feature vector for each translocation event. These flattened descriptive key names for the features were generated to maintain traceability in subsequent applications. All processed event sequences, their flattened features, and associated feature key names were saved as a Python pickle object for efficient storage and retrieval. A local peptide events database was employed to track these preprocessed pickle files, thereby preventing redundant segmentations and feature extractions from raw datasets.

## DL-based peptide classification

Peptide classification using DL models followed a common architectural schema: a sequence (either conductance states or raw current) was processed by a CNN or a TCN branch [40,41], the output of which was then concatenated with concurrently input event-level features (Figs 3A and S1C). This combined representation was subsequently fed into a fully connected (Dense) layer, culminating in a final softmax output for multiclass classification. Because experiments were conducted using single-analyte solutions (e.g., > 95% pure synthetic guest-host Ala peptide), the class labels for all translocation events are deterministic ground truths, not probabilistic estimates. This creates a rigorous supervised learning environment distinct from the unsupervised clustering required for unknown mixtures.

Three distinct DL models were evaluated. (i) The TCN-Dense (S + F) model was trained on the conductance state sequences (S) combined with event-level features (F). The architecture and hyperparameters for this model were identical to those previously described [22], with modifications only to the data loading from our current database. (ii) The CNN-Dense (S + F) model also utilized conductance state sequences (S) and event-level features (F). The sequence processing branch consisted of three one-dimensional CNN layers, each followed by a dropout layer with a rate of 0.3. The output of these CNN layers was then subjected to global max pooling before concatenation with the event-level features. The concatenated output was then passed to the final Dense classification layers. (iii) The CNN-Dense (C + F) model was structurally similar to the CNN-Dense (S + F) model but was trained on current sequences (C) instead of state sequences, combined with event-level features (F). Its sequence processing branch also employed three one-dimensional CNN layers

with dropouts of 0.3 and global max pooling prior to concatenation with the event-level features, followed by analogous Dense classification layers.

For all three models, data preparation and training procedures were standardized as follows. Data Balancing: during data loading, class balancing was performed *in situ* by downsampling all classes to match the number of events in the class with the fewest samples (S3 Table). Data Split: the balanced dataset was partitioned into an 80% training set and a 20% testing set. Feature Preprocessing: event-level features were scaled using a StandardScaler. Missing (NaN) values within the feature sets were imputed with a value of -1. Model Optimization: the Adam optimizer was used for model training, with categorical cross-entropy as the loss function. Model performance was monitored using metrics including accuracy, precision, and recall. Training Protocol: training was performed for approximately 100 epochs with a batch size of 32. To prevent overfitting and optimize training, several callbacks were implemented: early stopping (to halt training when validation performance no longer improved), learning rate reduction on plateau (to reduce the learning rate if validation loss stalled), and model checkpointing (to save the best performing model weights).

The classification performance of each trained model was subsequently evaluated on its respective unseen testing dataset. Comprehensive metrics, including overall accuracy, precision, recall, and F1-score for each peptide class, were summarized in a standard classification report. Additionally, confusion matrices were generated to provide a detailed visualization of per-class prediction accuracy and misclassification patterns.

### ML-based peptide classification

ML-based classification of peptide translocation events was performed using the gradient boosting framework, XGBoost [31]. For this, peptide event data, previously extracted and characterized into event-level features, were loaded from a local database. To ensure balanced class representation, all peptide classes were downsampled to match the class with the minimum number of events (S3 Table). The comprehensive dataset was then split into an 80% training set and a 20% testing set for model development and evaluation, respectively. For clarity, throughout this features-only model is referred to as XGBoost (F). The XGBoost classifier was configured with the following key parameters: a multiclass classification objective, the number of target classes set to the total number of peptides, 1000 boosting rounds (trees), and a learning rate of 0.05 to control the step size shrinkage. Regularization was applied with max depth of 5 to limit tree complexity, minimum child weight was 1 to control minimum sum of instance weight (hessian) needed in a child, gamma was set to 0 for minimum loss reduction required to make a further partition on a leaf node, subsample was 0.8 (fraction of samples used per tree), and the fraction of features used per tree was 0.8. L1 and L2 regularization were used. For reproducibility, a random state was fixed, and computation was distributed across all available CPU cores. The model's performance during training was monitored using the multiclass classification error. The trained model's performance was evaluated on the unseen testing dataset. Classification metrics including accuracy, precision, recall, and F1-score were summarized in a standard classification report, and a confusion matrix was generated to visualize per-class prediction accuracy.

### Hybrid DL/ML-based peptide classification

A hybrid classification approach, XGBoost (S + F), was developed to leverage the representational power of deep learning for sequence data alongside the robust performance of tree-based ensemble methods. This model combined sequence-derived embeddings from a supervised CNN with the previously defined event-level features, which were then merged and fed into the XGBoost classifier [31].

**CNN-based sequence embedding generation.** To generate sequence embeddings, a dedicated CNN model [40] was trained to perform multiclass peptide classification directly on conductance state sequences. Raw state sequences were first remapped to integer IDs, where original states (e.g., 0, 1, 2, 3) were shifted by one (e.g., 1, 2, 3, 4) to reserve '0' as a dedicated padding token. These remapped sequences were then post padded to a fixed sequence length of 1300 time points (which contained 99% of the events), thus ensuring a uniform input dimension for the CNN. During

data loading for CNN training, all peptide classes were downsampled to the size of the smallest class to maintain class balance. The dataset was subsequently split into an 80% training set and a 20% validation set.

The CNN embedding model architecture consisted of an embedding layer (input dimension vocab size was number of unique remapped states + 1 (to include padding token) and output dimension was 128) that also masked the padding token (value 0). This was followed by a stack of two 1D CNN layers, each with 128 filters and a kernel size of 3, utilizing rectified linear unit activation and 'same' padding. Each layer's output was subjected to batch normalization and max pooling with a pool size of 2, followed by a dropout layer of 0.4. A global max pooling layer then summarized the processed sequence into a fixed-size representation. This was fed into a Dense layer with an output embedding dimension of 128, producing the final sequence embedding. A separate Dense classification head (with softmax activation) was attached to this embedding layer, allowing the entire CNN to be trained in a supervised manner for peptide classification. The model was compiled using the Adam optimizer with sparse categorical cross-entropy for the classification head (the embedding output had loss of none as it was not directly optimized during this phase). Training was performed for 100 epochs with a batch size of 32, incorporating early stopping (restoring best weights) and reduce learning rate on plateau callbacks to prevent overfitting and optimize learning. The training history (loss and accuracy) were monitored on the validation set. After training, the final sequence embeddings were extracted from the trained model's embedding layer for both the training and testing sets.

**Hybrid model training and evaluation.** For the hybrid model, event data (including sequences and features) for all peptides were loaded from a local database, applying the same minimum event duration filter range as used for the individual DL and ML models (e.g., ≥ 20 ms was most optimal empirically). The dataset was then split 80–20 into training and testing sets. The extracted CNN sequence embeddings (from the previously trained encoder) were then horizontally concatenated with their corresponding event-level features for both the training and testing sets. This combined feature vector served as the input for the final XGBoost classifier. The XGBoost classifier was configured with identical parameters to those used when trained solely on event-level features (as detailed in the previous section). Early stopping was also applied during XGBoost training. The performance of the hybrid XGBoost (S + F) model was assessed on the independent test set using a comprehensive classification report, providing overall accuracy, precision, recall, and F1-score. A confusion matrix was also generated to visualize per-class prediction accuracy.

## Supporting information

**S1 Fig. Processing and analysis schemes. (A)** Data acquisition and state labeling. **(B)** Event segmentation and feature calculation. **(C)** ML/DL classification workflow.
(TIF)

**S2 Fig. PA nanopore stability enables capture of peptide translocation events.** Recordings of a single PA nanopore in absence (top) and presence of guest-host Ala peptide (bottom) at +70 mV potential showing relative stability of free nanopore. Peptide events were observed for guest-host Ala peptide at 20 nM concentration. Recordings made from the same membrane and the same single nanopore. Data sampled at 400 Hz. Standard buffer conditions were symmetrical 100 mM KCl, 20 mM succinate, pH 5.6.
(TIF)

**S3 Fig. DL-based classification confusion matrices plotted with absolute event counts.** CNN-Dense (C + F) model classification results for two extreme minimum event duration filters (5 ms, left; 20 ms, right) where event counts are plotted in the two confusion matrices. The corresponding normalized confusion matrices are given in the main text in Fig 3B and 3C.
(TIF)

**S4 Fig. Tree-based classification confusion matrix with absolute event counts.** Confusion matrix for XGBoost classification using the event-level feature set showing absolute counts in the class-balanced test data set. This matrix represents the best-performing XGBoost (F) model at a minimum event duration of 20 ms. The normalized confusion matrix corresponding to these absolute count data is in the main text in Fig 4.
(TIF)

**S1 Table. Recording time per peptide class in the dataset.** These recording times include all data at the 70 mV voltage condition and peptide concentration range (5–20 nM).
(DOCX)

**S2 Table. Feature engineering definitions.** A total of 69 biophysical features were extracted for each translocation event based on the 4-state kinetic model (States 0, 1, 2, 3) observed in the PA nanopore. Features corresponding to states or transitions not observed in a specific event are assigned NaN, which the XGBoost algorithm natively handles as informative signals.
(DOCX)

**S3 Table. Translocation event counts at different minimum event duration filters.** Counts of events before and after class-balancing downsampling for both the 5 ms and 20 ms minimum event duration datasets.
(DOCX)

**S4 Table. Model performance metrics at varying minimum event duration.** Performance based on test set evaluation metrics (Accuracy, Precision, Recall, F1-score) of different ML/DL classification models at minimum event durations ranging from 5 ms to 20 ms (N = 1 scan). Finalized replicated performance metrics (N = 5) are presented in Table 1.
(DOCX)

## Acknowledgments

We thank members of the department as well as Tobin Sosnick for useful feedback and discussions. Portions of this document, including some of the Python code and language refinement, were generated with the assistance of AI-powered tools. All content was reviewed and approved by the authors, who take full responsibility for its accuracy.

## Author contributions

**Conceptualization:** Jennifer M Colby, Bryan A. Krantz.

**Data curation:** Jennifer M Colby, Bryan A. Krantz.

**Formal analysis:** Bryan A. Krantz.

**Funding acquisition:** Bryan A. Krantz.

**Investigation:** Jennifer M Colby, Bryan A. Krantz.

**Methodology:** Jennifer M Colby, Bryan A. Krantz.

**Project administration:** Bryan A. Krantz.

**Resources:** Bryan A. Krantz.

**Software:** Bryan A. Krantz.

**Supervision:** Bryan A. Krantz.

**Validation:** Bryan A. Krantz.

**Visualization:** Bryan A. Krantz.

**Writing – original draft:** Jennifer M Colby, Bryan A. Krantz.

**Writing – review & editing:** Jennifer M Colby, Bryan A. Krantz.

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
