## [Decision Letter · Decision Letter 0]

5 Feb 2026

PCOMPBIOL-D-25-02595

A Dynamical Anthrax Toxin Nanopore Biosensor for High-Fidelity Single-Peptide Classification

PLOS Computational Biology

Dear Dr. Krantz,

Thank you for submitting your manuscript to PLOS Computational Biology. After careful consideration, we feel that it has merit but does not fully meet PLOS Computational Biology's publication criteria as it currently stands. Therefore, we invite you to submit a revised version of the manuscript that addresses the points raised during the review process.

We look forward to receiving your revised manuscript.

Kind regards,

Eduardo Jardón-Valadez

Academic Editor

PLOS Computational Biology

Nir Ben-Tal

Section Editor

PLOS Computational Biology

**Journal Requirements:**

4) Please amend your detailed Financial Disclosure statement. This is published with the article. It must therefore be completed in full sentences and contain the exact wording you wish to be published.

**Reviewers' comments:**

Reviewer's Responses to Questions

**Comments to the Authors:**

Reviewer #1: General comments:

Novelty check:

1. PUBMED:

https://pubmed.ncbi.nlm.nih.gov/?term=A+Dynamical+Anthrax+Toxin+Nanopore+Biosensor+for+High-Fidelity+Single-Peptide+Classification

No observable hit at all

2. Dimensions.ai (with ERA 2023, Norwegian register level 1/2, and Pubmed database enabled). Registration to this database is free:

https://app.dimensions.ai/discover/publication?search_mode=content&or_facet_publication_type=article&or_facet_journal_list=ERA%202023&or_facet_journal_list=Norwegian%20register%20level%201&or_facet_journal_list=Norwegian%20register%20level%202&or_facet_journal_list=PubMed&search_text=A%20Dynamical%20Anthrax%20Toxin%20Nanopore%20Biosensor%20for%20High-Fidelity%20Single-Peptide%20Classification&search_type=kws&search_field=full_search

Available hits point out to the papers by the authors of this manuscript

Hereby, the topic is uniquely developed by the authors.

The manuscript deals with a niche, which is a nanopore biosensor for anthrax toxin. It is thoroughly written, and all indicators and parameters pertaining the computational methods are all elaborated accordingly.

Specific comments:

- line 99: Don't put figure in the intro section. Move it to result and discussion instead

- Line 374-377: What would be a translational or clinical setting that would require optimization of the four-state peptides? There is concerns that ML- enhanched translocation signals could be noisy.

Kindly elaborate to strengthen your biological narrative.

- Line 429-433: Why did you pick KKKKKXXSXX as the peptide motif for the general sequence?

As I checked in the references, the terminal polylysine provide strong positive charge for good electrophoretic capture. Moreover, in ML-perspective, it could minimize confounders.

Although the reference is there, kindly elaborate the justification!

- Line 460-463: You have different states of event detection here, peptide-bound, fully blocked, and open. The depiction of the states actually fit well to be processed with Hidden Markov Models (HMM). Why not use HMM instead? HMM works great on transition probabilities and decoding state path with Viterbi algorithms.

There are some papers in pubmed that doing so:

https://pubmed.ncbi.nlm.nih.gov/?term=hidden+markov+model+protein+annotation

Note that those papers did not belong to my, my research group, nor my collaborators.

- Methodology section: Please kindly provide at least a reference citation to each paragraph of your methodology section!

- Line 480: What is your consideration to select range of 5 to 20 ms? Is there any prior computational works with this threshold?

Reviewer #2: Review of A Dynamical Anthrax Toxin Nanopore Biosensor for High-Fidelity Single-Peptide Sequencing

Summary. This paper describes the beginning of a very exciting advance in nanopore sequencing. Single molecule sequencing that takes advantage of the inherent properties of the anthrax protein transduction pore.

1) Do the polylysine portions of the test sequence provide a special advantage in these proof-of-principle experiments? The authors should discuss in detail how the physical properties of this highly polar, highly soluble, and highly cationic test peptide helped them achieve success in this demonstration. Does the polylysine respond to the TM gradient? Is this interaction necessary for sequencing?

2) The authors should discuss how difficult it could be to get sequence information from peptides with other physical chemistries. Zwitterionic, hydrophobic, anionic, prone to helicity.

3) The authors should discuss in more detail how multiple reads of the same sequence could be combined to generate more accurate sequence information.

**Have the authors made all data and (if applicable) computational code underlying the findings in their manuscript fully available?**

Reviewer #1: Yes

Reviewer #2: Yes

PLOS authors have the option to publish the peer review history of their article (what does this mean? ). If published, this will include your full peer review and any attached files.

**Do you want your identity to be public for this peer review?** For information about this choice, including consent withdrawal, please see our Privacy Policy .

Reviewer #1: No

Reviewer #2: No

**Figure resubmission:**
---

## [Decision Letter · Decision Letter 1]

13 Feb 2026

Dear Dr. Krantz,

We are pleased to inform you that your manuscript 'A Dynamical Anthrax Toxin Nanopore Biosensor for High-Fidelity Single-Peptide Classification' has been provisionally accepted for publication in PLOS Computational Biology.

Best regards,

Eduardo Jardón-Valadez

Academic Editor

PLOS Computational Biology

Nir Ben-Tal

Section Editor

PLOS Computational Biology

Reviewer's Responses to Questions

**Comments to the Authors:**

Reviewer #1: The authors have revised the manuscript as instructed. Therefore, I accept this manuscript for publication.

Reviewer #2: The authors have done an excellent job revising the manuscript, which was already novel and very interesting.

**Have the authors made all data and (if applicable) computational code underlying the findings in their manuscript fully available?**

Reviewer #1: Yes

Reviewer #2: None

PLOS authors have the option to publish the peer review history of their article (what does this mean? ). If published, this will include your full peer review and any attached files.

**Do you want your identity to be public for this peer review?** For information about this choice, including consent withdrawal, please see our Privacy Policy .

Reviewer #1: No

Reviewer #2: No

---

## [Editor Report · Acceptance letter]

PCOMPBIOL-D-25-02595R1

A Dynamical Anthrax Toxin Nanopore Biosensor for High-Fidelity Single-Peptide Classification

Dear Dr Krantz,

I am pleased to inform you that your manuscript has been formally accepted for publication in PLOS Computational Biology. Your manuscript is now with our production department and you will be notified of the publication date in due course.

With kind regards,

Anita Estes
